# Interruptions in a dental setting and evaluating the efficacy of an intervention: A pilot study

**Carsten Ziegler, Pratik J. Parikh** *

Department of Industrial Engineering, University of Louisville, Louisville, KY, United States of America

* pratik.parikh@louisville.edu

## Abstract

### Introduction

Interruptions during dental treatment are frequent, and often impact provider satisfaction and processing times We investigate the source and duration of such interruptions at a German dental clinic.

### Methods

A pre-post approach was adopted at this dental clinic. This included direct observations of 3 dentists and 3 dental hygienists, and a survey of providers. Following that, an intervention (switchable 'Do Not Enter' sign) was chosen, and a pilot study was conducted to evaluate if the chosen intervention can reduce processing time and improve provider satisfaction. Additional observations and surveys were performed afterwards.

### Results

Pre-intervention data indicated that interruptions have the highest negative impact on provider satisfaction at this clinic as well as on processing time during longer and more complex treatments, where a minor error due to an interruption could lead to rework of 30 minutes and more. The total number of interruptions dropped by 72.5% after the intervention, short interruptions (< 1min) by 86%. Provider survey indicated improvement due to the intervention in perceived workload, provider work satisfaction, patient safety and stress.

### Conclusions

This study demonstrates that a switchable sign can substantially reduce the number of interruptions in this dental clinic. It also shows the potential of improving the work environment by reducing interruptions to the dental providers.

**Data Availability Statement:** The data cannot be publicly shared due to ethical or legal restrictions. For this pilot study, data was collected from a small group of participants (6 providers) at a dental clinic in suburban Berlin, Germany. Information such as

this, when combined with the patient load, type of equipment at this clinic, and number of dentists and hygienists mentioned in the Methods section, may risk the identification of study participants. Qualified researchers may request access to the data through the University of Louisville's Institutional Data Access/Ethics Committee. Access to confidential data will be granted to researchers who meet the criteria established by the committee. For inquiries regarding data access and ethical considerations, please contact: Christy LaDuke, MA, CIP, CCRP Director, Human Research Protections University of Louisville Louisville, KY, USA.

**Funding:** The author(s) received no specific funding for this work.

**Competing interests:** The authors have declared that no competing interests exist.

## 1. Background

While constant innovations in dentistry and dental technologies (e.g., application of lasers, complex root canal treatments involving microscopes, iPad-controlled handpieces and expensive new filling materials, and the constantly evolving dental chairside CAD/CAM system) attempt to provide definitive dental care, they can be cost prohibitive. In Germany, most of these technologies are only available to private / self-pay patients or those that make a significant financial contribution of their own beyond the public healthcare system's coverage, which only spends 6% of its 2020 budget on dental care [1]. These patients do not only expect quality care on a technical level ('what is delivered'), but also value how their treatments are delivered regarding their safety, comfort, and well-being.

While medical errors in general pose a major threat to patient safety, with estimates reporting between 210,000 and 440,000 preventable deaths annually in the U.S. [2], others estimate that medical errors could be the third leading cause of death in the U.S. [3]. An error in dentistry is certainly less likely to cause death, but it could still lead to potentially painful and expensive additional treatments. In 2019 a study done by the German public health insurance system (MDK) [4] showed, that 14,533 cases of possible medical malpractice had to be reviewed by experts, of which 1,055 cases (8.4%) were related to dentistry. Out of these a total of 392 cases (37.2% of the reviewed cases) turned out to be malpractice. Broken down further, 384 cases (134 of which had errors) were related to dental caries treatment and 308 cases (120 errors) were endodontics related. These statistics only cover major errors that were brought to the attention of a review board; the actual numbers–especially of smaller errors–are naturally higher. A study on implementation of an error reporting system suggested underreporting in health care close to 90% [5].

Interruptions in a dental clinic are frequent and can contribute to errors in dentistry. By interruptions, we mean anything external that takes a dental providers' attention away from a task or communication activity they are engaged in a part of their work (including interruptions caused by dental assistants). While working as a sales director for several Tier 1 Ag/Construction OEM suppliers, one of the co-authors had the opportunity to help build and grow a dental clinic in Germany. Interruptions at this clinic, especially during critical steps of longer treatments, often caused major losses of time (30 minutes and more). This could lead to missed dental caries during examination, preparation, fillings or dental chairside CAD/CAM manufactured parts falling out due to wetness or other mistakes during the application of adhesives, to name a few. All of them would likely result in unnecessary loss of more tooth substance and discomfort for the patient, or in rare cases, turn out as life threatening, if these parts fall out while the patient is asleep, and they might aspirate them. These instances generate waste for everyone as the patient would need to spend extra time either at once (should the problem be noticed right away) or during another appointment. In either case, the practice would need to redo the treatment, spending precious time and costly dental materials.

While there is a growing body of research involving interruptions in healthcare, in general, we are not aware of any research that pertains to the specific challenges faced by dental practitioners due to interruptions to their treatments. While we knew prior to this study that interruptions were affecting operations at this dental clinic, we neither knew what caused these interruptions nor what type of intervention to design and implement. Our objective in this exploratory, not hypothesis-driven, study was to address the following questions:

Q1. What are the types of interruptions during dental treatments in this clinic?

Q2. What is the impact of interruptions on the dental treatment (e.g., wait times and provider satisfaction) in this clinic?

Q3. What intervention(s) would help mitigate the most detrimental interruptions in this clinic?

To address these questions, we used actual data collected at a German dental clinic to analyze the interruptions and their impact on two quantitative outcome measures: processing time and provider satisfaction survey. Based on this, we devised an intervention and piloted it at this dental clinic.

## 2. Methods

Our approach used direct observation to collect quantitative data to categorize interruptions to understand the impact on lost time and provider survey data to understand their impact on provider satisfaction. The study was approved by the University of Louisville's Institutional Review Board as a quality improvement project.

### 2.1. Setting

Due to regulations in Germany, only dentists are the key providers in German dental offices. They are allowed to delegate certain treatments to specially trained staff (like hygienists) that still need to work under their supervision (even though that supervision does not need to be constant or immediate). All work done by hygienists is the responsibility of the delegating dentist, while the practice owner is responsible for the work done by any employed dentists. Dental assistants are limited to performing tasks (on a patient) under direct supervision of a dentist, and their primary focus during a treatment is on preparing and handing tools and materials, use of suction to keep the treated area clean and dry, and to use the curing light (for polymerization). If time permits, they also handle the documentation on the computer.

The data for this study was collected at a dental clinic in a suburb of Berlin. The clinic was open 5 days a week, 12 hours per day and had 6 treatment rooms. It was staffed with 3 dentists, 3 hygienists (at different qualification levels), a receptionist and 8 assistants, who assisted dentists during the treatment or perform other tasks such as lab work, cleaning / disinfection, or sterilization procedures. Due to regulations, at any given time at least one of the dentists needed to be present in the clinic during operating hours.

The annual patient load was approximately 4,000 patients across all age groups. The patient load was spread almost evenly across the year. The dental equipment included 3 dental CAD/CAM computers and 3 milling machines, dental microscope, 3D cone beam, and multiple dental lasers. The clinic had originally been set up without phones in treatment rooms or an intercom system to avoid interruptions and distractions during treatments.

Every dentist's treatment was performed by a team of one dentist and one or two dental assistants, who prepared the required materials and tools before the treatment, took care of the suction during the treatment, kept the area around the treated teeth clear and accessible and handed all necessary tools and materials to the dentist as required. After the patient had left, the assistant took care of cleaning and disinfecting the workplace including the patient's seat.

Hygienist usually worked alone, with their appointments lasting either 60 minutes (adults) or 30 minutes (children). Dentists' appointments (aside from check-ups and other minor tasks) were scheduled starting at 30 minutes with added 15-minute increments, with the longest treatments carrying on for up to 3 hr. Except for emergencies, all treatments are scheduled in advance; there were no walk-ins. Depending on X-ray and/or visual inspection, a preferred treatment (composite vs. inlay/crown) was determined, and a time slot was allotted according to the expected size and complication of the cavity. This time slot did not leave any room for

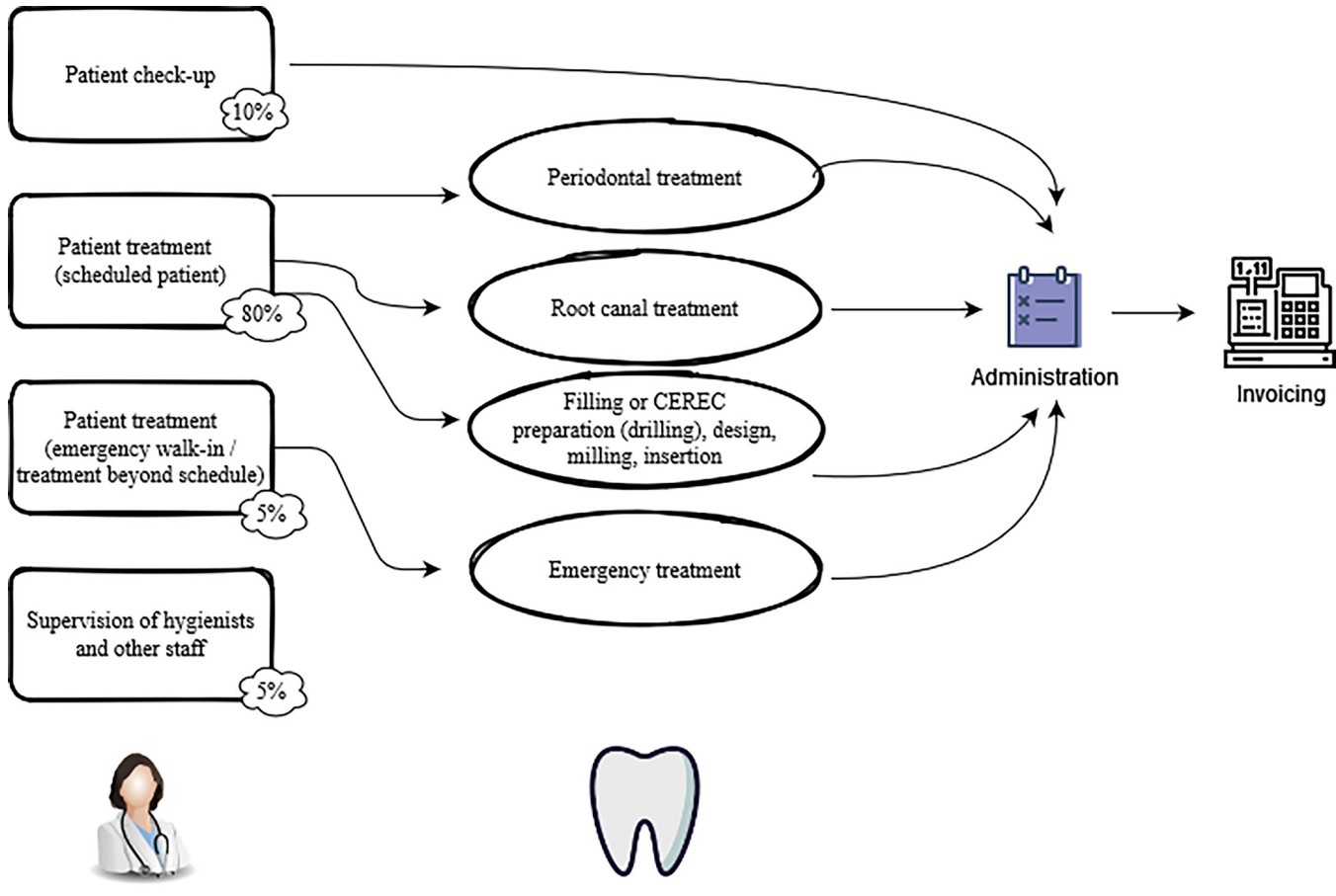

**Fig 1. Key activities of a dentist.**

error or any 'major' interruption. Deviations from the schedule resulted either in overtime or potentially treatments that needed to be cancelled.

A somewhat simplified view of the standard procedures in Fig 1 illustrates the three key areas of treatment at this clinic that were observed to be sensitive to interruptions: periodontic treatment, root canal treatments, and dental chairside CAD/CAM inlays/crowns. This also has an immediate negative impact on provider satisfaction as having to 'unnecessarily' repeat the most critical steps due to an interruption also adds stress and workload, especially when repeating a critical step is associated with a large time loss (e.g., dental chairside CAD/CAM rework or having to redo a root canal). Table 1 summarizes the various types of interruptions and their sources considered in our study.

## 2.2. Participants

The clinic's 3 dentists and 3 hygienists took part voluntarily in direct observation and all responded voluntarily to a 25-question survey. The providers were between 25 and 55 years old (average 40) and had anywhere between 5 and 25 years of professional experience in their assigned positions (average 11.5 years). The participants were enrolled upon obtaining verbal consent (recorded by the first author) and had the opportunity to opt in or out of the study.

Table 1. Definitions of sources of interruptions assessed during dental treatment.

| Source | Definition |
|---|---|
| Provider | Interruptions resulting from other providers in the practice |
| Dental Assistants | Interruptions caused by assistants directly assisting the observed provider |
| Receptionist | Interruptions by the receptionist (authorization, phone call, etc.) |
| Staff | Interruptions caused by dental assistants or other staff members not currently assisting the observed treatment |
| Other patient | Interruptions by a patient not currently attended to |
| Emergency | Occurrences of an emergency (medical) |
| Technical malfunction | Interruptions due to equipment failure |
| Other | Miscellaneous interruptions not assigned to any other category |

## 2.3. Data collection

**2.3.1. Pre-intervention period.** Two types of data were collected; processing time and satisfaction survey.

For the processing time, the dentists and hygienists were observed during all hours of the workday (8 am—8 pm) and all five days of the workweek to capture the nature of interruptions. A total of 100 hours of direct observation was conducted during March 7, 2022—August 26, 2022. We adapted an observation data form based on existing literature [6–9]. Data recorded for each interruption included task interrupted, a description of the interrupting event, location, source, medium, and time. The event description included reason for the interruption (task request, receive info, provide info, or authorize/sign a document) and whether relocation or change of task was required. Free form fields were used to record any observed impact of the interruption. To provide context for observed interruptions, we also noted the times and task categories the dentist/hygienist engaged in while being observed. Any interruption that had providers move their eyes away from the treated area was considered dissatisfactory due to the eye strain caused by significantly changing light brightness and focus.

All providers in the clinic were observed and data samples collected to classify the source of interruption, the task interrupted and the medium of interruption, while recording the day, time, and the duration of the interruption as well. Provider assistants were also included (wherever they were relevant, either as a part or the cause of interruption). Special attention was given to critical procedures and treatments with major impact on the care provision. In general, anything that had a significantly higher chance of harming the patient or creating disproportionate rework effort was considered critical. Patients in the clinic were not interviewed, nor was their care affected as the data collection is purely observational.

After enrolling the participant (dentist/hygienist), the observer (lead author) shadowed them noting the time when the participant changed tasks and capturing data from observed interruptions. The observer followed without verbal interaction except when on the first patient contact during the observation session, at which time the dentist would ask the patient for permission to have the observer watch the treatment. No patients declined to allow the observer to watch their treatment and no patient information was collected.

For the provider satisfaction data, we designed a 25-question survey. The purpose of this survey was to capture how interruptions were viewed by dentists and hygienists during dental treatments. Topics included how interruptions impact daily workload, time / wait time, patient safety, and care provider stress, as well as their perceived impact on patients. Additionally, questions about how interruptions occur, and the techniques used by providers to manage interruptions were include. Participation was voluntary with each participant receiving a sheet

of paper presenting the opportunity to anonymously complete the survey. In the survey instructions, interruptions were defined as "anything that takes your attention away from a task or communication activity that you were already engaged in as part of your job."

**2.3.2. Post-intervention period.** We used the same observation data form as used during the pre-study for the quantitative data. During November 1, 2022 –January 6, 2023, another 100 hours of direct observation was conducted. While we used the same 25-question survey used during the pre-intervention period, we added a few more questions that were directly related to the provider experience of using the intervention.

## 2.4. Intervention

Our chosen intervention (upon discussions with the providers) was a flashing 'Do not enter' sign mounted above the entrance door of each treatment room (see Fig 2). This light could be turned ON to alert staff and others indicating that a critical procedure is in progress which should not be interrupted (Fig 3)

A remote control was connected to each sign to turn the sign ON and OFF, and was installed in each room within easy reach of the provider / assistant team in each room (Fig 4). Each remote control was placed into a plastic bag to ensure easy disinfection without damage to the devices. Each staff member was trained to understand the meaning of the sign, with special focus on the providers, to ensure the lights were turned on during procedures they had identified as 'critical' during the pre-intervention period.

After installation and training, a trial and ramp-up period of 4 weeks was used to acclimatize the providers on the new procedure. All staff members were given the opportunity to ask questions and report any issues during the 4-week period. Following these 4 weeks, the post-intervention data collection was conducted.

To illustrate how the intervention was used, consider as an example treatment: preparing and filling a tooth with a composite filling (simplified procedure shown in Fig 5). The light bulbs indicate when and for which approximate duration the sign would have been turned on.

## 3. Results

### 3.1. Pre-intervention period

During this period, a total of 194 interruptions were observed over 100 hours of data collection. At that time, the clinic did not have any formal policies for protecting providers from interruptions during this period.

Fig 6 shows a summary of observations by source of interruption. Most interruptions occurred because people entered a treatment room from the outside to have their issues addressed, not worrying about the consequences their interrupting actions may have on the ongoing treatment.

Nearly 30% (58/194) of these interruptions involved the provider providing information to others, 15% (30/194) receiving information, 25% (48/194) involved a task request, and 15% (30/194) required the dentist to authorize / sign a document. The remaining 15% (28/194) fell in other categories (mainly technical issues). Over 20% (40/194) of observed interruptions occurred during critical tasks. Overall, 30% (59/194) of interruptions caused the providers / assistants to relocate.

On average, providers were interrupted every 31 minutes (i.e., ~2 interruptions per hour). Distributed according to the daily workload, most interruptions occurred while more providers were working simultaneously and during normal office hours (Fig 7). This was expected, as the higher patient throughput and larger number of providers working at the same time

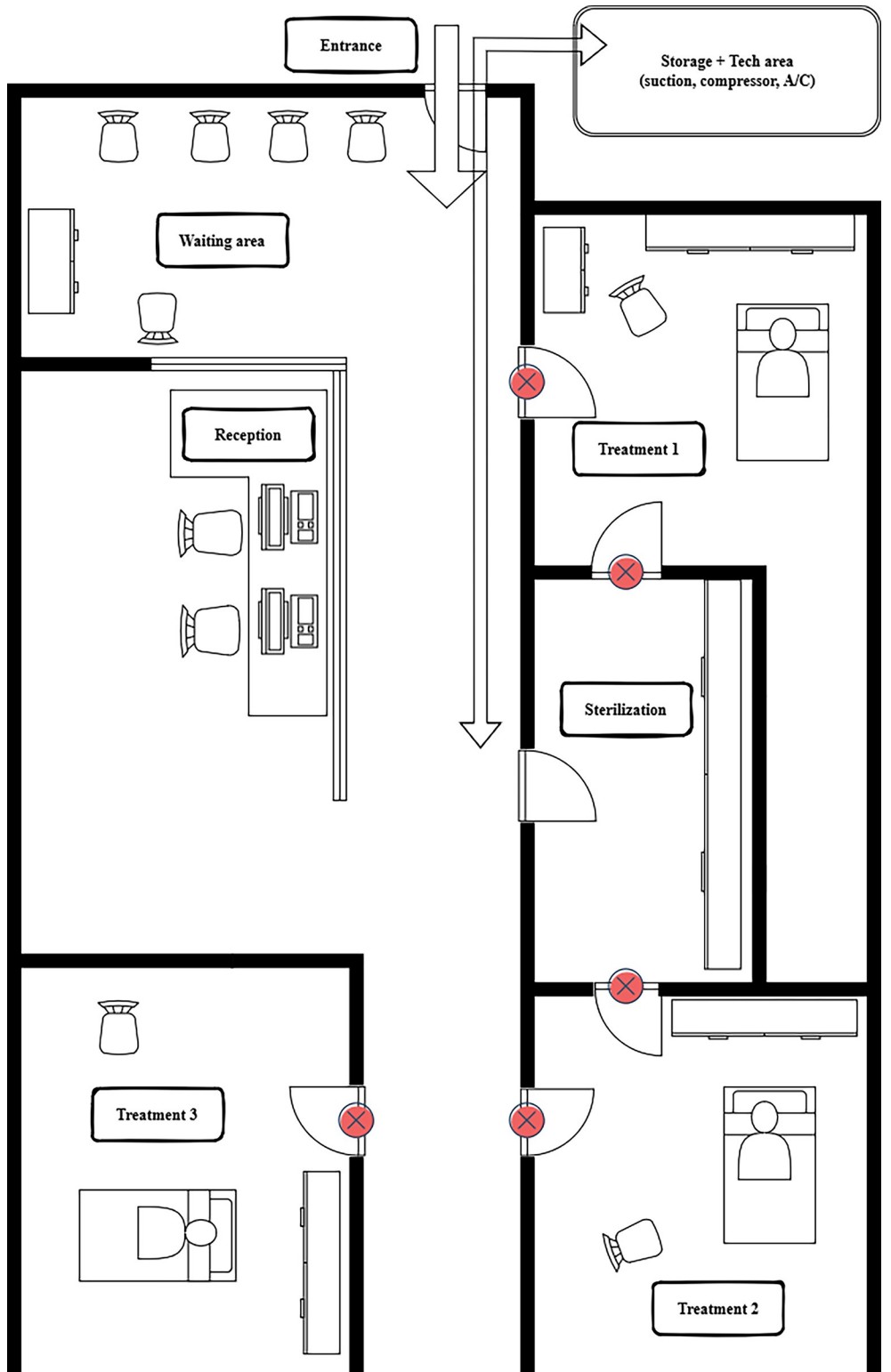

**Fig 2. Layout of the rooms with positions of the signs (X).**

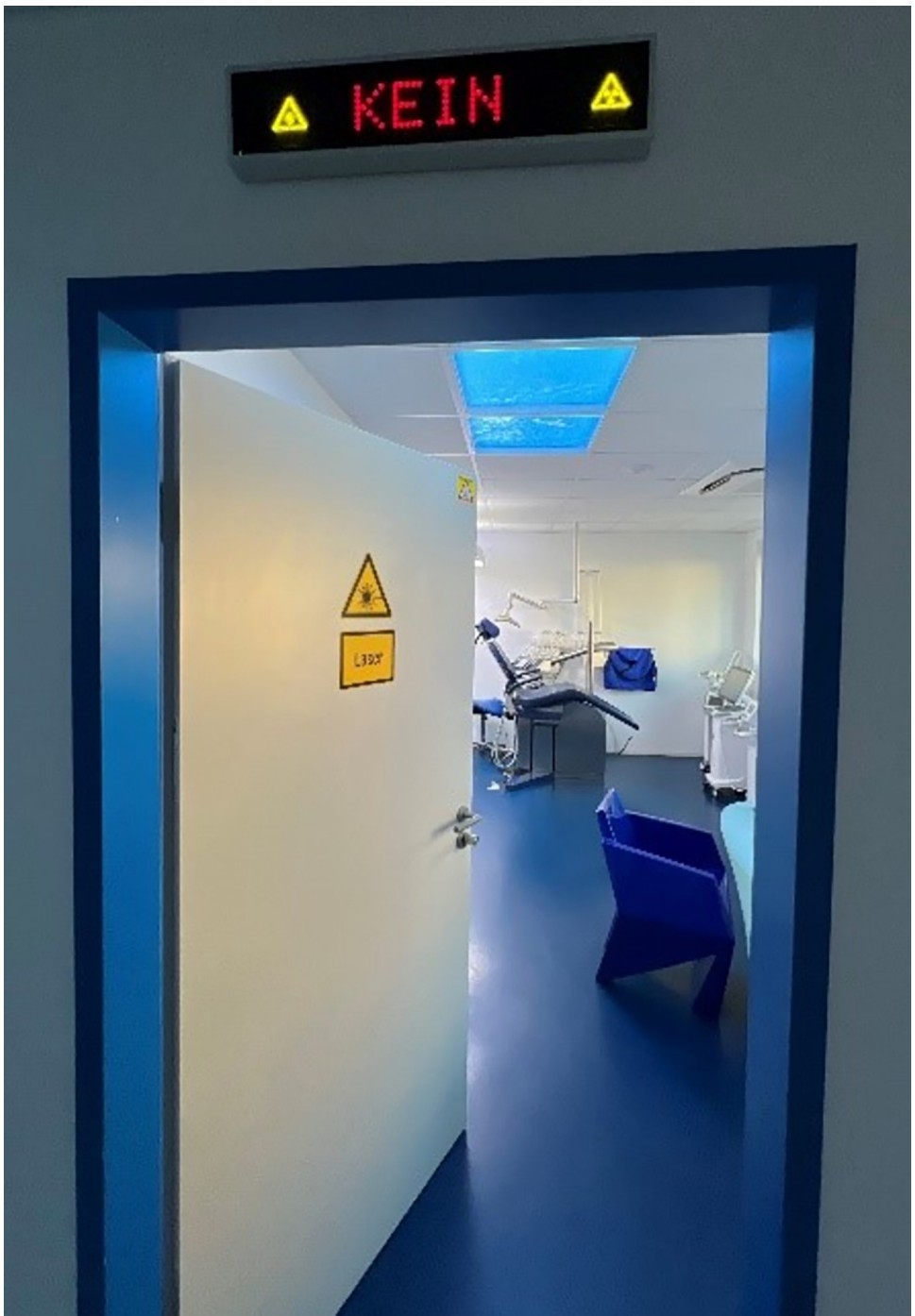

**Fig 3. Sign above door (Kein in German refers to No or Do not Enter).**

should result in more interactions between staff members as well as more external interactions.

The duration of the observed interruptions varied greatly; either they were rather short (23% were under 1 minute) and consisted of a brief verbal exchange of information or tended to be time consuming (39% took longer than 10 minutes to resolve). These usually either

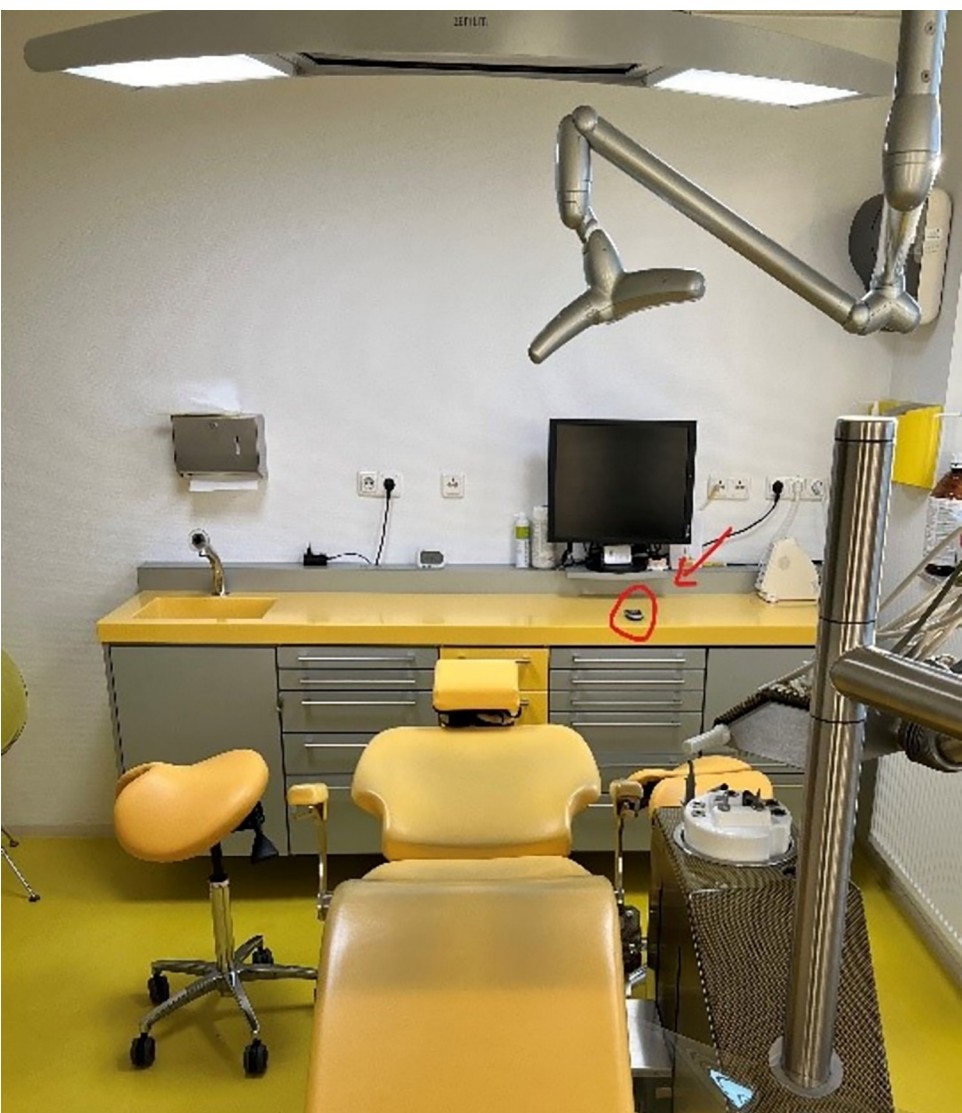

**Fig 4. Position of the remote control.**

involved a provider having to work on someone or something else in between, doing some rework because of the interruption, or technical issues (Fig 8).

In terms of provider satisfaction survey, while 83% of the providers agreed that interruptions lead to noticeable time losses / increased wait times, none supported the idea that all interruptions should be eliminated. All providers agreed that interruptions should be eliminated during pre-defined critical procedures (such as work close to apex, bonding, and

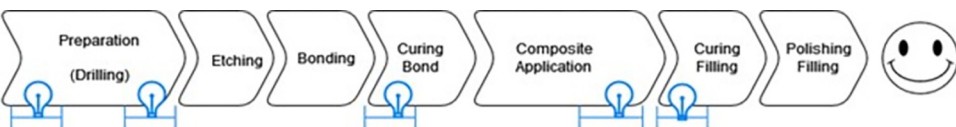

**Fig 5. Illustration of the time periods the intervention was used during a sample treatment.**

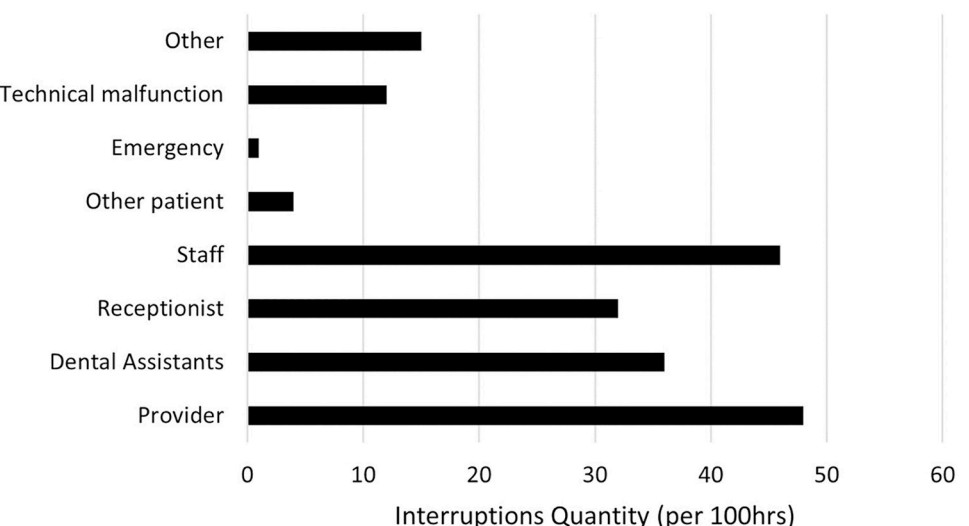

**Fig 6. Sources of interruption.**

adhesive steps to name a few). Further, everyone agreed or strongly agreed that interruptions have a negative impact on their work satisfaction (Table 2).

## 3.2. Post-intervention period

During this period, a total of 53 interruptions were observed over another 100 hours of data collection. The clinic used the 'do not enter' signs to protect providers from interruptions during this period.

Fig 9 shows a summary of observations by source of interruption. Most of the remaining interruptions (about 45%; 24 out of 53) occurred because the dental assistants working on the patient made mistakes or did not properly prepare their work. This was followed by technical malfunctions, which remained unchanged compared to the initial observation.

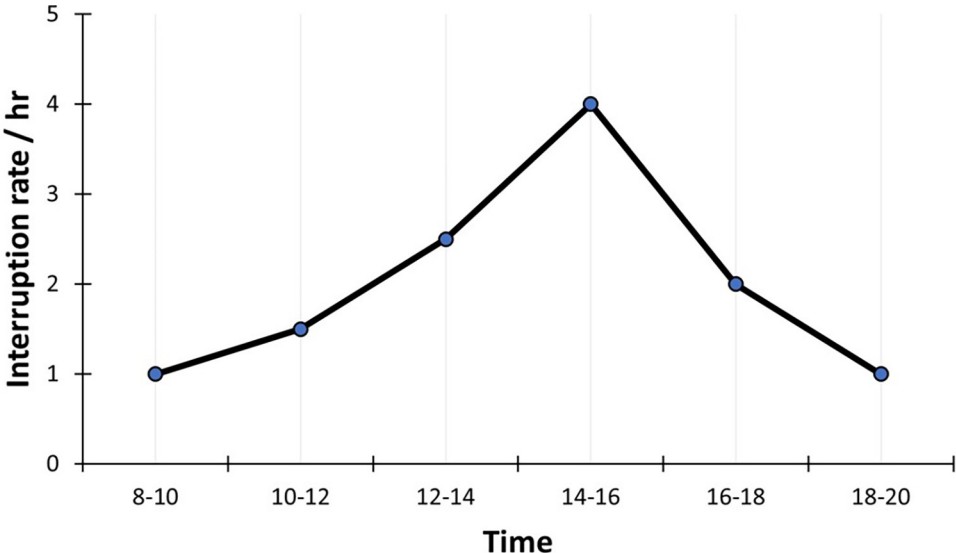

**Fig 7. Interruption rate.**

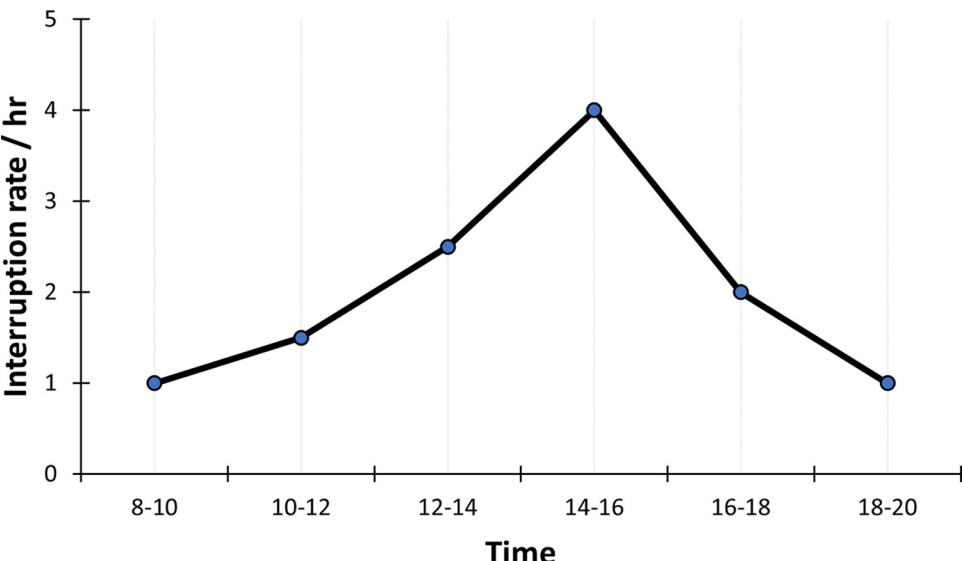

**Fig 8. Duration of interruptions.**

The duration of the observed interruptions was significantly reduced as well. Fig 10 shows a comparison between the observation pre and post intervention. For instance, shorter interruptions (<1 min) reduced from 44 to 6 (86% reduction) while longer interruptions (10 minutes and over) reduced from 76 (57+19) to 28 (18+10), a 63% reduction. The number of interruptions for a given duration observed during post-intervention were significantly smaller than those observed during pre-intervention (p = 0.015; Wilcoxson Signed Rank Test).

Related to the satisfaction survey, Table 3 summarizes the providers' responses to the post intervention survey and compares them to the initial responses before they were given the ability to choose when not to be interrupted from the outside. After the intervention, providers experienced interruptions as less of a burden to their daily workload, did not feel they led to as much time loss, had less negative impact on their work satisfaction, and caused less negative stress. These results are also reflected in Table 4 where an improvement of the providers' work satisfaction was observed, together with a perception of better work quality and patient safety.

To ensure the intervention did not create unanticipated problems itself, questions were asked relating to the usability and user experience of the intervention (see Table 5). Across all

**Table 2. Provider survey related to interruptions they experienced.**

| Interruptions experienced while working on your patients. . . | Strongly Disagree | Disagree | Neutral | Agree | Strongly agree |
|---|---|---|---|---|---|
| . . . add to your daily workload | | | 2 | 2 | 2 |
| . . . lead to noticeable time loss / increased wait | | | | 5 | 1 |
| . . . have negative impact on your work satisfaction | | | | 3 | 3 |
| . . . could place your patients at risk | | | | 5 | 1 |
| . . . are sometimes beneficial | | 1 | 4 | 1 | |
| . . . cause you additional (negative) stress | | | | 4 | 2 |
| . . . should be eliminated during critical procedures | | | | | 6 |
| . . . should ALL be eliminated | 1 | 5 | | | |

Techniques to manage interruptions included waiting to turn toward an interrupting person until they were finished with the current task or taking notice of an interrupting person but asking them to wait for a few moments or to leave documents behind on a desk to be dealt with later.

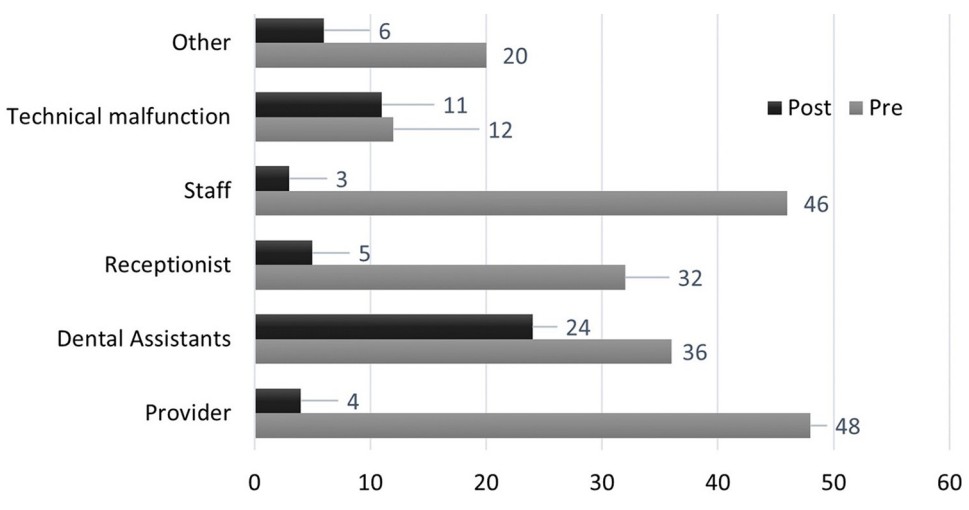

**Fig 9. Interruptions by source.**

questions, the providers responded as having a positive impact on their workflow with minimal interruptions.

## 4. Discussion

To our knowledge, this study was the first to evaluate interruptions at a single dental clinic, along with an implementation of an intervention. Quantitative data collected from observations at this clinic suggested that the observed interruptions tend to break the delivery of steady treatment. Further, a satisfaction survey from the providers suggested their work satisfaction degenerated due to frequent interruptions, which required them to refocus, caused them stress and increased the risk of having to do rework. A simple, yet easy-to-understand, and sustainable intervention such as the 'Do not enter' sign used in our study at this clinic can drastically

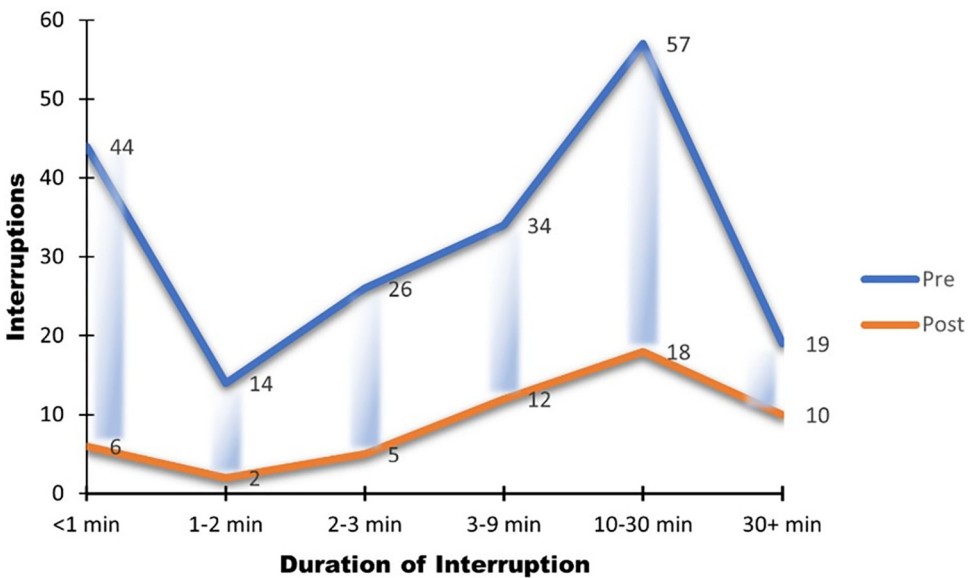

**Fig 10. Reduced duration of interruptions.**

**Table 3. Providers' responses on their experiences with interruptions while working on their patients pre and post intervention on a Likert scale; 1 –strongly disagree to 5 –strongly agree.**

| | Average Response | | |
|---|---|---|---|
| Interruptions experienced while working on your patients. . . | Pre | Post | Improvement |
| . . . add to your daily workload | 4.0 | 3.0 | Yes |
| . . . lead to noticeable time loss / increased wait | 4.2 | 2.5 | Yes |
| . . . have negative impact on your work satisfaction | 4.5 | 2.8 | Yes |
| . . . could place your patients at risk | 4.2 | 2.7 | Yes |
| . . . cause you additional (negative) stress | 4.3 | 2.8 | Yes |
| . . . should be eliminated during critical procedures | 5.0 | 2.7 | Yes |

reduce unwanted interruptions and improve provider satisfaction, potentially leading to improved dental care.

Several studies have focused on the rate of interruptions and their effects. In a German hospital emergency department, Weigl et al. [10] noted 7.5 interruptions per hour, while it was 5.3 interruptions per hour in an internal medicine and surgery ward of a German hospital [11]. For a U.S. hospital ICU, approximately 3 interruptions per hour were observed [12, 13]. In our study, we found interruptions at our specific dental clinic to occur twice per hour, which, although smaller than the other settings, was still perceived as bothersome by the affected dental providers. While many interruptions were <1 min (23%), a small number of them (10%) were rather long (>30 min). A short interruption period did not guarantee that no mistake was made, but none was noticed at the time and the procedure restarted quickly. But in case of a long interruption that was caused due to an error, depending on the treatment performed and the process step it was in, it could lead to time loss due to rework of 30 minutes and more. Examples of this included an inlay or crown improperly attached, a tooth broke during the insertion, and adhesive not properly cured.

When it comes to interruption sources, they vary among the various specialties based on specific characteristics of those specialties [14–16]. A recent study at French hospital units summarized various interrupters from the point of view of work functions involved [17]. Unlike what some studies in the hospital nursing environment suggested [18–20], the majority (75%; 145/194) of the interruptions we observed in our dental clinic were not self-induced by the staff members, but these interruptions occurred due to sources who entered the treatment rooms from the outside as indicated in Fig 6. This corroborates with previous studies, which also suggested that other health care team members are the largest source of interruptions [21–24]. Family members were not a source of interruptions during treatment as they were generally not allowed to be in the treatment room at our dental clinic. In our setting, we further noticed that most interruption sources did not have the urgency to justify an immediate interruption to a treatment process. Upon deeper look at the sources of interruptions, we noticed

**Table 4. Providers' self-assessment of different aspects of their work quality on a Likert scale: 1 –strongly disagree to 5 –strongly agree.**

| Post intervention . . . | Average Response | Improvement |
|---|---|---|
| . . . the work environment improved patient safety | 4.0 | Yes |
| . . . the quality of my work improved | 4.0 | Yes |
| . . . the perceived time lost decreased | 4.5 | Yes |
| . . . I felt less stressed | 4.3 | Yes |
| . . . my overall work satisfaction improved | 4.7 | Yes |

**Table 5. Providers' opinions about using the intervention in their daily work on a Likert scale; 1 –strongly disagree to 5 –strongly agree.**

| Using the intervention . . . | Average Response | Change |
|---|---|---|
| . . . I experienced problems operating the sign | 1.0 | positive |
| . . . I used the sign during critical procedures | 4.5 | positive |
| . . . the extra steps added noticeable workload | 2.0 | positive |
| . . . caused new interruptions that wouldn't have occurred | 1.0 | positive |
| . . . helped mitigate/eliminate the effects of the most prevalent interruptions in the daily work | 4.7 | positive |
| . . . eliminated interruptions during critical procedures | 4.2 | positive |

that many of them were related to paperwork or requests for information or obtaining supplies or equipment, all of which could have waited a few minutes to avoid interrupting a critical task.

Interruptions can also have detrimental effects on job satisfaction, workflow efficiency and patient safety. Prior studies suggest that they could be an important factor in the medical providers' workload and could result in medication as well as other errors [15, 25–28]. Our survey at this dental clinic found that interruptions add to the providers' daily workload and cause them additional stress, as highlighted with scores of 4.0 and 4.3 out of 5 in Table 3. Clearly, dental providers at this clinic need to have a sharp focus on a variety of complex and potentially harmful (to the patient) tasks. Any interruption that breaks this focus not only adds workload due to the additional work to be performed by the providers at this clinic, but also adds stress as the providers need to refocus, knowing that the likelihood of making an error has just been increased due to the fact their original procedure had been interrupted. While this finding is limited to our dental clinic, delay in refocusing on the primary task and potential errors have also been observed in lab studies and a study with emergency physicians [29, 30].

The satisfaction survey data further confirmed that dental provider satisfaction at this clinic degraded with interruptions, especially if they were involved in critical processes and/or result in lengthy rework procedures. This is again indicated in Table 3 by an average response of 4.5 on a Likert scale, agreeing that interruptions had a negative impact on work satisfaction. While our finding is limited to this clinic, other studies have indicated that as staff turnover presents a serious challenge to health care [31], work satisfaction is an essential factor in medical staff retention.

From an intervention standpoint, based on whether or not the intervention is attached to the provider or otherwise moved together with them throughout their work environment, we can categorize them as 'dynamic' or 'static' interventions. Examples of 'dynamic' interventions include drug round tabards [32], vests [22], and red disposable aprons [18]. However, due to the 'static' nature of the services provided at this dental clinic, our intervention of 'Do not enter' light was selected. We created a static, protective bubble around the treatment area, which blocked the dynamics of the outside world during critical phases of the treatment. Having an item attached to a dental provider would not have been a suitable solution as it would not have been noticeable before someone opened the door to the treatment room, which by itself has the potential to cause an interruption.

The impact of our intervention at this clinic was the creation of a quieter and, thus, a more structured treatment environment. Our survey data showed that our intervention helped mitigate the effects of the most prevalent interruptions in the providers' daily work (strong agreement of 4.7 on Likert scale–Table 5). While our finding is limited to this clinic, other studies had alluded to similar observations; e.g., the 'sterile cockpit' in the aviation industry that allows

pilots to perform their tasks without interruptions during defined critical periods, leading to a lower risk of errors [20, 33]. Our data showed a strong (4.2 on Likert scale) elimination of interruptions during critical procedures at this clinic (Table 5).

We were able to achieve a reduction in short interruptions (<1min) of 86% and 63% for longer interruptions (>10 min) at this clinic, with most providers adhering to the proper use of the sign and staff members observing it. We reduced the number of 'avoidable' interruptions (those that could potentially be eliminated by the intervention) by 87.5%, from 146 to 18 interruptions over the 100 hr observation period (Figs 9 and 10). This differs from findings of Federwisch et al. [34] who determined one of the reasons for failure of their intervention of a sterile cockpit approach to be lack of compliance. The ease of use and the convincing results from our study at this clinic suggest that other clinics may be able to adapt this intervention as a sustainable, long-term solution.

We also observed that providers found their modified work environment improved patient safety and the quality of their work (both 4.0 on Likert scale, Table 4), resulting in a strong improvement of the overall work satisfaction (4.67 on the Likert scale, Table 5). While this is specific to our dental clinic, Bell et al. [25] show that medical staff, in general, feel that reducing interruptions improves patient safety, workload, accuracy, job satisfaction and mental health.

Retaining qualified staff members is critical today, as finding a suitable replacement can take weeks or months, thus costing tens of thousands of dollars and endangering patient satisfaction should they not be treated in a timely fashion due to staff shortage. While at our practice, we already had a system that needed modification to convert it to a 'Do not enter' sign, a $1,000 per room cost to set up such a sign with a remote appears quite reasonable compared to the loss in revenue to the clinic, and patient satisfaction and eventual loyalty.

There are, however, limitations of this study. First, the sample size was small (6 providers) and was performed within a mid-size German dental practice. Hence, our findings may not be generalizable to other practices of different sizes or locations, as their system of care, layout, and staffing pattern may be different, and cultural factors might affect the impact of the interventions in other countries/settings as well. However, our study does provide some understanding interruptions at our dental practice, with a possible easy-to-implement and sustain intervention. We recommend that our approach be used (with appropriate modifications) for other practices to evaluate interruptions and test our proposed intervention. Second, the pre-post nature of the study, unlike a randomized controlled trial, could have allowed confounding factors to positively or negatively impact findings at this clinic. However, utmost care was taken by the lead author to ensure no other parallel process improvement efforts were in place or any other procedural changes occurred during the pre- and post-periods. Third, analysis of the case mix of patients and how it varied from pre to post study were not considered. Patient case mix may influence the number and nature of interruptions in multiple ways; e.g., length of treatment or the requirement for authorization on documents for other patients. However, given the sufficiently long period of data collection (100+ hours during pre- and post-periods), variations in the daily case mix would have averaged out. Finally, it is unknown to what extent the observer influenced the behavior of staff during the assessment. Efforts were made to limit the Hawthorne effect by the observer maintaining a distance from the dental unit/treatment area, adopting a discreet observation behavior and not engaging in conversation with staff.

Future work at this dental clinic could include a timer that would allow for the interrupting provider to see the remaining time on the 'Do not enter' sign and return at a later time. This could help improve overall efficiency, as the person wanting to enter may decide to either wait in front of the door a short period of time or choose to continue on other tasks and return later. Replicating this study at other dental clinics, of different sizes and in different countries, would help generalize findings derived from our study at this single dental clinic.

## Author Contributions

**Conceptualization:** Carsten Ziegler, Pratik J. Parikh.

**Data curation:** Carsten Ziegler.

**Formal analysis:** Carsten Ziegler, Pratik J. Parikh.

**Investigation:** Carsten Ziegler, Pratik J. Parikh.

**Methodology:** Carsten Ziegler, Pratik J. Parikh.

**Project administration:** Pratik J. Parikh.

**Supervision:** Pratik J. Parikh.

**Validation:** Pratik J. Parikh.

**Visualization:** Carsten Ziegler.

**Writing – original draft:** Carsten Ziegler, Pratik J. Parikh.

**Writing – review & editing:** Carsten Ziegler, Pratik J. Parikh.

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
