## [Decision Letter · Decision Letter 0]

25 Sep 2023

PONE-D-23-23150An Observational Study of Interruptions in a Dental Setting and Evaluating the Efficacy of an InterventionPLOS ONE

Dear Dr. Parikh,

Thank you for submitting your manuscript to PLOS ONE. After careful consideration, we feel that it has merit but does not fully meet PLOS ONE’s publication criteria as it currently stands. Therefore, we invite you to submit a revised version of the manuscript that addresses the points raised during the review process. Please submit your revised manuscript by Nov 09 2023 11:59PM. If you will need more time than this to complete your revisions, please reply to this message or contact the journal office at plosone@plos.org. Please include the following items when submitting your revised manuscript:A rebuttal letter that responds to each point raised by the academic editor and reviewer(s). You should upload this letter as a separate file labeled 'Response to Reviewers'.A marked-up copy of your manuscript that highlights changes made to the original version. You should upload this as a separate file labeled 'Revised Manuscript with Track Changes'.An unmarked version of your revised paper without tracked changes. You should upload this as a separate file labeled 'Manuscript'.If applicable, we recommend that you deposit your laboratory protocols in protocols.io to enhance the reproducibility of your results. Protocols.io assigns your protocol its own identifier (DOI) so that it can be cited independently in the future. For instructions see: https://journals.plos.org/plosone/s/submission-guidelines#loc-laboratory-protocols. Additionally, PLOS ONE offers an option for publishing peer-reviewed Lab Protocol articles, which describe protocols hosted on protocols.io. Read more information on sharing protocols at https://plos.org/protocols?utm_medium=editorial-email&utm_source=authorletters&utm_campaign=protocols.

We look forward to receiving your revised manuscript.

Kind regards,

Naveed Sadiq, Ph.D.

Academic Editor

PLOS ONE

Journal Requirements:

2. Please ensure that you include a title page within your main document. You should list all authors and all affiliations as per our author instructions and clearly indicate the corresponding author.

3. Please include a caption for figure 8.

4. Please ensure that you refer to Figure 8 in your text as, if accepted, production will need this reference to link the reader to the figure.

5. We note you have included a table to which you do not refer in the text of your manuscript. Please ensure that you refer to Table 2 in your text; if accepted, production will need this reference to link the reader to the Table.

**Additional Editor Comments:**

Please address all the comments of reviewers, especially of reviewer 1 and resubmit the revised manuscript.

Reviewers' comments:

Reviewer's Responses to Questions

**Comments to the Author**

1. Is the manuscript technically sound, and do the data support the conclusions?

Reviewer #1: Yes

Reviewer #2: Yes

Reviewer #3: Yes

2. Has the statistical analysis been performed appropriately and rigorously? 

Reviewer #1: No

Reviewer #2: I Don't Know

Reviewer #3: No

3. Have the authors made all data underlying the findings in their manuscript fully available?

Reviewer #1: No

Reviewer #2: No

Reviewer #3: No

4. Is the manuscript presented in an intelligible fashion and written in standard English?

Reviewer #1: Yes

Reviewer #2: Yes

Reviewer #3: Yes

5. Review Comments to the Author

Reviewer #1: The article discuss the interruptions during dental procedures and its impact on the provider. The research question wasn't stated clearly in a sense what is the hypothesis? It is causal that when interruptions occur during any given procedure will lead to delays and negative impacts. using the switchable signs reduced the interruptions as the statistics showed, however, the technical factors were missing from the collected data, for example What time of the day most of the interruptions happening, What type of procedure was conducting (intensity of the procedure and the focus varies). Plus other interruptions like calls/intercom from the reception weren't measured. The research paper measured on type of interruptin and tested the switch sign neglecting other interruptions variable in the dental clinic setting.

Reviewer #2: Thank you for this interesting, pertinent and well-written paper.

Several comments related to the methods (data analysis):

I am interested to know why a test of significance was not applied, applying a relevant test will make the findings observed differences) more meaningful. For instance, the authors may look for the mean differences in the duration of the interruptions before and after. I see that authors have shown the differences the duration in figure 10.However it would be worthwhile to apply an appropriate test if significance, rather than only looking at differences.

Regarding the ‘qualitative data’. Will it be appropriate to call a survey with likert scale responses as qualitative data? Did the authors employ qualitative data collection methods? The findings/providers responses from the ‘survey’ have been quantified (table 2, 3 and 4). I would suggest rephrasing this heading and avoid using the term ‘qualitative data’.

It will be useful to clearly define ‘critical procedures’ for the readers.

Minor comments:

Below are some of the typos but the whole manuscript needs to be proof read.(particularly for tenses)

Please spell out (CEREC) in the introduction.

P.2. To address these questions, we use actual data collected at a German dental clinic…. (To address these questions, we ‘used’ actual data collected at a German dental clinic

P.5. The remaining 15% (28/194) fall in other categories (mainly technical issues). (‘fell’ in other category?)

Reviewer #3: This is an interesting study.

The sample size limitation dictates that the findings/results cannot be generalized.

I would suggest adding, A pilot study to the title.

References should be completed and as per guidelines to the authors

6. PLOS authors have the option to publish the peer review history of their article (what does this mean?). If published, this will include your full peer review and any attached files.

Reviewer #1: No

Reviewer #2: No

Reviewer #3: No

---

## [Author Response · Author response to Decision Letter 0]

8 Nov 2023

Response letter has been attached.

---

## [Decision Letter · Decision Letter 1]

18 Dec 2023

Interruptions in a Dental Setting and Evaluating the Efficacy of an Intervention: A Pilot Study

PONE-D-23-23150R1

Dear Dr. Parikh,

We’re pleased to inform you that your manuscript has been judged scientifically suitable for publication and will be formally accepted for publication once it meets all outstanding technical requirements.

Kind regards,

Naveed Sadiq, Ph.D.

Academic Editor

PLOS ONE

Additional Editor Comments (optional):

Thank you very much for addressing all the concerns.

Reviewers' comments:

Reviewer's Responses to Questions

**Comments to the Author**

1. If the authors have adequately addressed your comments raised in a previous round of review and you feel that this manuscript is now acceptable for publication, you may indicate that here to bypass the “Comments to the Author” section, enter your conflict of interest statement in the “Confidential to Editor” section, and submit your "Accept" recommendation.

Reviewer #2: All comments have been addressed

Reviewer #3: All comments have been addressed

2. Is the manuscript technically sound, and do the data support the conclusions?

Reviewer #2: Yes

Reviewer #3: Yes

3. Has the statistical analysis been performed appropriately and rigorously? 

Reviewer #2: Yes

Reviewer #3: Yes

4. Have the authors made all data underlying the findings in their manuscript fully available?

Reviewer #2: No

Reviewer #3: No

5. Is the manuscript presented in an intelligible fashion and written in standard English?

Reviewer #2: Yes

Reviewer #3: Yes

6. Review Comments to the Author

Reviewer #2: The comments made have been addressed.

1. Wilcoxon test has now been applied to analyse interruptions.

2. 'Qualitative data' has been changed by the authors to quantitative survey.

3. Details about 'critical procedures' have now been added to the methods section.

Reviewer #3: Thanks for the revised manuscript.

There are two points to be addressed:

1. Author's citation in the manuscript text should be harmonious with author's name, without initials. In the text, there is a lack of consistency in this regard. Somewhere, authors name with INITIAL letters and at some areas without Initials. There should be consistency in the style.

2. Out of 33 references, many of them lack consistent reference writing style. Please note, References # 1, 3, 4, 5, 7, 8, 11, 12, 13, 14, 17, 22, 26, 27, 30 and 33. They should be consistently written in reference style as per guidelines to the authors for the journal.

7. PLOS authors have the option to publish the peer review history of their article (what does this mean?). If published, this will include your full peer review and any attached files.

Reviewer #2: No

Reviewer #3: **Yes: **Khalid Almas

---

## [Editor Report · Acceptance letter]

16 Mar 2024

PONE-D-23-23150R1 

PLOS ONE

Dear Dr. Parikh, 

I'm pleased to inform you that your manuscript has been deemed suitable for publication in PLOS ONE. Congratulations! Your manuscript is now being handed over to our production team.

Kind regards, 

on behalf of

Dr. Naveed Sadiq 

Academic Editor

PLOS ONE